# Remote Patient Monitoring Is Associated with Improved Outcomes in Hypertension: A Large, Retrospective, Cohort Analysis

**DOI:** 10.3390/healthcare12161583

**Published:** 2024-08-09

**Authors:** Wesley Smith, Brett M. Colbert, Tariq Namouz, Dean Caven, Joseph A. Ewing, Andrew W. Albano

**Affiliations:** 1HealthSnap, Miami, FL 33136, USA; wes@healthsnap.io (W.S.); bmc48@miami.edu (B.M.C.); 2Prisma Health Upstate, Greenville, SC 29605, USA; tariq.namouz2@prismahealth.org (T.N.); alex.ewing@prismahealth.org (J.A.E.); 3Virginia Cardiovascular Specialists, Mechanicsville, VA 23116, USA; dcaven@vacardio.com

**Keywords:** remote patient monitoring, hypertension, telehealth, chronic care management

## Abstract

Hypertension (HTN) is a chronic condition that requires careful monitoring and management. Blood pressure readings in the clinic and self-reported blood pressure readings are often too intermittent to allow for careful management. Remote patient monitoring is a solution that may have positive impacts on HTN management. Individuals at cardiac and primary care clinics were prescribed a remote patient-monitoring (RPM) program. Patients were sent blood pressure monitors that were enabled to transmit data over cellular networks. We reviewed trends in HTN management retrospectively in patients who had previously been on conventional therapy for a year and participated in RPM for a minimum of 90 days. There were 6595 patients enrolled, and the mean duration on RPM was 289 days. A total of 4370 participants (66.3%) had uncontrolled HTN, and 2476 (37.5%) had stage 2 HTN. After at least 90 days on the RPM program, the number of patients with uncontrolled HTN reduced to 2648 (40.2%, *p* < 0.01), and the number of patients with stage 2 HTN reduced to 1261 (19.1%, *p* < 0.01). Systolic blood pressure improved by 7.3 mmHg for all patients and 16.7 mmHg for stage 2 HTN. There was improvement in mean arterial pressure (MAP) in all patients with uncontrolled HTN by 8.5 mmHg (*p* < 0.0001). RPM is associated with improved HTN control and provides further evidence supporting telehealth programs which can aid in chronic disease management.

## 1. Introduction

Remote patient monitoring (RPM) is a system to track patient vital signs and health metrics outside traditional healthcare settings [1]. This can include using wirelessly enabled blood pressure cuffs, thermometers, pulse oximeters, scales, and glucometers to transmit vitals to a provider from the patient’s home on a regular basis. Integrating RPM into chronic disease management can potentially improve patient quality of life and mitigate healthcare costs by enabling the timely detection of health deterioration, thereby reducing unnecessary emergency department visits and hospital admissions [2,3].

Hypertension (HTN), despite its growing prevalence, often remains undiagnosed [4,5]. Conventionally, patients diagnosed with HTN were required to visit their healthcare provider regularly for in-office blood pressure monitoring and treatment. However, this method may be affected by “White Coat Syndrome” and “Masked Uncontrolled HTN”, where office measurements may not accurately reflect the patient’s average blood pressure in a home or ambulatory setting [6]. Given the extended intervals between in-office assessments, uncontrolled or poorly managed blood pressure is common [7]. To address this issue, patients can monitor their blood pressure remotely, which involves the patient recording their home blood pressure measurements and submitting their readings to their clinician’s practice or aggregating readings and providing them during a scheduled appointment [8].

The conventional method of self-monitoring blood pressure at home, complemented with regular counseling and education, has been evaluated in a meta-analysis by Tucker et al. The study, which incorporated data from 25 trials and 10,487 patients, documented a moderate yet statistically significant decrease in systolic pressure by 3.2 points and a reduction in diastolic pressure by 1.5 points [8].

Recently, there has been an increase in RPM usage in place of traditional self-report measurements. This change has stemmed from new Centers for Medicare and Medicaid Services (CMS) billing codes introduced in 2019 and 2020 [9]. Despite the growth in patient and provider participation, there are few published reports of clinical outcomes associated with RPM. Emerging research suggests RPM may have an effect on the management of HTN, with the potential for it to become a standard method for monitoring blood pressure with greater accuracy. However, verifying the efficacy of RPM for long-term blood pressure control requires further study [10,11,12]. This study investigates the association between RPM and changes in blood pressure in the HTN population.

## 2. Materials and Methods

This was an observational, retrospective, cohort analysis of 6595 patients. The cohort was drawn from patients at the primary care services of a large healthcare organization (Prisma Health Upstate, Greenville, SC, USA) and a cardiology practice (Virginia Cardiovascular Specialists, Mechanicsville, VA, USA) who were enrolled in an RPM program provided by HealthSnap, Inc. (Miami, FL, USA). Patients were included in the cohort if they participated in an RPM program at either institution.

Individuals were referred for RPM based on medical necessity, as determined by the patients’ healthcare providers. Indications included HTN, atrial fibrillation, atrial flutter, myocardial infarction, stroke, transient ischemic attack, ischemic heart disease, non-ischemic heart disease, heart failure, and chronic kidney disease. Participants had received at least a year of conventional treatment for HTN without RPM prior to enrollment in RPM.

Upon enrolling in the RPM program, prescribed patients received a cellular-enabled, FDA-cleared, digital, automatic blood pressure monitor from HealthSnap, Inc. This device automatically transmitted data to a HIPAA-compliant remote patient portal. A team of HealthSnap-licensed care navigators (registered nurses) monitored this portal, guiding patients on how to correctly measure their blood pressure according to the American Heart Association’s recommended protocol [13]. Briefly, patients were advised to follow certain pre-measurement steps, including resting quietly; avoiding caffeine, exercise, or smoking; emptying their bladder; and wearing comfortable clothing. They were also educated on the correct sitting position and cuff placement for accurate readings. To account for variations and confirm reading accuracy, two consecutive measurements were taken for each participant, with a 1–2 min interval between each measurement. Participants could take their blood pressure readings at any time of day and multiple times per day. To account for time and frequency variability, the entire week’s readings were averaged together to obtain the weekly value that we used for our analysis.

If a patient’s blood pressure fell outside of goal range or had not been transmitted for more than three consecutive days, care navigators would contact them. Monthly meetings were held between patients and care navigators to discuss care plan goals approved by their physicians, focusing on managing their condition and maintaining healthy blood pressure thresholds. Clinical staff consultations involved evidence-based lifestyle strategies for managing blood pressure, such as dietary changes, regular physical activity, stress management, and sleep hygiene. Clinical staff also provided notes that integrated with the EHR to support necessary provider feedback.

Data were pulled from the RPM’s secure platform in a de-identified manner. For inclusion, participants had to be enrolled in RPM for at least 90 days and transmit blood pressure data through the wirelessly enabled blood pressure cuff for at least 30% of the days. This was to ensure that effects could be correlated to RPM use. To account for correct use of the blood pressure cuff, the baseline and endpoint data were the average of the first 7 days of data transmission for the baseline and the average of the last 7 days of data transmission for the endpoint.

The patients were categorized as 1. Controlled/Uncontrolled HTN (systolic blood pressure was ≥130 mmHg, or their diastolic pressure was ≥80 mmHg) [14] and 2. stage 2 Systolic HTN (systolic blood pressure was ≥140 mmHg, or if their diastolic pressure was ≥90 mmHg [15]).

In addition to exploring the overall correlations of the RPM program participation to HTN management, we stratified the participants into quartiles of duration of RPM defined as 90–154 days, 155–221 days, 222–365 days, and any duration beyond 365 days. Each analysis was performed independently, with mean arterial pressure (MAP) serving as the primary outcome measure. The aim of these additional analyses was to determine whether longer program participation correlated with improved MAP.

The information analyzed in this study was devoid of any personal patient identifiers. The findings were presented as collective averages rather than individual patient results.

## 3. Results

### 3.1. Overall Participation

A total of 6595 patients were included in this analysis, with an average participation duration of 295 days (±182 days) and a data transmission index of 69.8% (±18.9%). 

### 3.2. RPM Is Associated with Improvements in Blood Pressure Control

In our total sample of 6595 patients who were in the RPM program, we found that 4370 patients (66.3%) had uncontrolled blood pressure based on the average their first seven measurements. After an average duration of 289 days on the RPM program, the number of patients with uncontrolled HTN decreased to 2648 (−18.3%, *p* < 0.01; Figure 1A,B).

A total of 2476 patients (38%) had stage 2 HTN at baseline. After an average duration of 290 days on the RPM program, this number fell to 1261 (−19%, *p* < 0.01; Figure 1C,D).

### 3.3. RPM Is Associated with Changes in Blood Pressure Readings

The baseline systolic pressure for the entire cohort of 6595 patients was 135 mmHg and improved by 7.3 points to reach an average of 127.8 mmHg.

In the uncontrolled HTN, the average systolic blood pressure improved from 143.5 mmHg at baseline to 131.6 mmHg after RPM (−11.9 mmHg, *p* < 0.0001). In the stage 2 HTN subset, the systolic pressure improved from 151.6 mmHg at baseline to 134.9 mmHg after RPM (−16.7 mmHg, *p* < 0.0001, Figure 2A).

In our patient pool, we observed a correlation of RPM use with improvements in diastolic pressure (Figure 2B). The overall average baseline diastolic pressure of 78.8 mmHg improved from 4.4 mmHg (*p* < 0.0001) to 74.4 mmHg. Notably, patients initially presenting with uncontrolled HTN had improvement, with an average diastolic pressure reduction from 82.3 mmHg at baseline to 76.0 mmHg. Furthermore, the subset of patients with stage 2 HTN experienced a reduction of 9 mmHg, with their average diastolic pressure decreasing from 85.5 mmHg to 76.5 mmHg. 

The baseline pulse pressure (PP) for the entire population was 56.3 mmHg. We observed an improvement of 2.8 mmHg, reducing the average PP to 53.4 mmHg. In patients initially presenting with uncontrolled HTN, the PP changed from 60.8 mmHg to 55.6 mmHg, an improvement of 5.1 mmHg. Moreover, in the subgroup of patients with stage 2 HTN, we saw a decrease in PP from a baseline of 66.1 mmHg to 58.4 mmHg (*p* < 0.0001; Figure 2C).

### 3.4. Longer RPM Participation Is Associated with Improved Outcomes

We assessed the correlation of the duration of RPM use with changes in the mean arterial pressure (MAP). We analyzed the changes in MAP among 4370 patients with uncontrolled HTN (Figure 3A). We observed an improvement in MAP of 8.5 mmHg (*p* < 0.0001). Subsequently, we segmented the program duration into quartiles based on the number of days (Figure 3B). The quartiles were defined as follows: the first quartile represented 90–154 days in the program; the second was 155–221 days; the third comprised 222–365 days; and any duration beyond 365 days fell into the fourth quartile. As the duration increased, we observed a correlation with improvements in MAP. Specifically, the second, third, and fourth quartiles showed a significantly greater reduction in MAP compared to the first quartile (*p* < 0.0001; Kruskal–Wallis Test), suggesting a positive correlation between the program’s duration and its usefulness in managing HTN.

## 4. Discussion

HTN is identified by consistent BP readings of ≥140/90 mmHg and is a significant risk factor for cardiovascular disease. This condition afflicts nearly 116 million American adults, yet only about 21% are effectively managed [16]. In 2019, HTN was associated with over 516,000 deaths in the U.S. alone. The prevention and control of HTN can yield notable health benefits. Studies have shown that even a minor reduction in the SBP can result in fewer cases of heart failure, coronary artery disease, and stroke [17].

Historically, the provision of healthcare services has been predominantly confined to institutional settings such as clinics and hospitals, necessitated by specialized, costly, and non-portable diagnostic and monitoring tools. However, the past two decades have borne witness to substantial advancements in medical device technology and internet accessibility, promoting a paradigm shift in healthcare delivery [18,19]. RPM exemplifies this evolution by enabling healthcare providers to remotely monitor and manage chronic conditions such as diabetes, HTN, heart failure, and obesity. Despite these advancements, it is noteworthy that the adoption of technology in healthcare has lagged compared to other industries, thereby highlighting an opportunity for further enhancement in efficiency and efficacy within the sector [20,21].

The COVID-19 pandemic underscored the necessity for telehealth services and RPM, as both patients and providers found themselves navigating a fundamentally altered healthcare landscape [22,23]. Over the past five years, there were abundant industry assertions positing the effectiveness of RPM for HTN management. However, these claims necessitate a more rigorous scientific evaluation to determine whether there is evidence substantiating the safety and effectiveness of RPM programs, and to comprehend their broader impact.

The regular interaction with care navigators and the availability of data for providers to inform clinical adjustments could explain why the RPM program yields superior results compared to self-monitoring at home without the support of a digital health platform and care navigator system. Future research should aim to more accurately identify this threshold and explore the factors contributing to this saturation effect, providing valuable insights for optimizing the use of RPM treatment for uncontrolled HTN.

This study sought to assess the association of RPM with improvements in HTN management. The findings from our study add to the existing, albeit limited, body of evidence supporting the use of RPM in diverse healthcare settings. We saw an association of RPM use with improvements in SBP, DBP, PP, and MAP across all users and especially those with uncontrolled or stage 2 HTN.

### 4.1. Treatment Duration Is Associated with Improved HTN Control

Our study underscores the crucial finding that the effectiveness of the RPM program in managing uncontrolled HTN is associated with increased duration on the RPM program, with patients experiencing significant benefits for at least a year or longer. As we segmented the program duration into quartiles, a trend emerged showing progressive improvements in MAP. Specifically, patients who participated in the program for 155–221 days (Q2), 222–365 days (Q3), and beyond 365 days (Q4) achieved greater reductions in MAP compared to those who were involved for only 90–154 days (Q1). This highlights that sustained patient engagement in the RPM program over an extended period may be instrumental in successfully managing HTN. Therefore, future strategies should prioritize maintaining long-term patient involvement in the RPM program.

### 4.2. Limitations of the Present Study and Future Directions

This study was a retrospective cohort analysis. Although participants had been on conventional HTN treatment for a year prior to baseline, there was no control or comparative group. The improvements seen in the cohort are likely due to a multitude of factors, including medication use, lifestyle modifications, and RPM use. As such, we were not able to quantify the effect that RPM has in isolation. However, our study provided data on real-world usage in a very large cohort (*n* = 6595).

A recent meta-analysis demonstrated that even a modest 5 mmHg reduction in SBP could lead to a 10% decrease in serious cardiovascular events [24]. This fact highlights the significant potential for cost savings derived from this RPM approach. Research suggests that RPM for HTN is more cost-effective compared to standard care [25,26,27]. Moreover, the substantial improvements in blood pressure observed in patients in our study would likely lead to significant reductions in events, utilization, and costs. For instance, stage 2 hypertensive patients could potentially see a 40% reduction in cardiovascular events based on the effects of the blood pressure improvements, thus underscoring the potential of these results to reduce costs and improve HTN management [24,28]. Future research should include healthcare utilization endpoints as well as subjective assessments of participant quality of life. There are ongoing studies that prospectively randomize participants to RPM or a control condition that could distinguish the effects of RMP on HTN control from the other factors listed above. There are ongoing studies that prospectively randomize participants to RPM or a control condition that could distinguish the effects of RPM on HTN control from the other factors listed above.

## 5. Conclusions

This large study provides a compelling association between RPM and improved blood pressure management when integrated with provider EHRs and coupled with regular counseling designed to enhance patient self-efficacy and health literacy. These improvements not only suggest potential reductions in HTN-related health events, comorbidities, and mortality, but also underscore the need for further research in these areas. 

Furthermore, these findings indicate that the improvements may increase with time over the course of the program, which lasted up to and beyond a year. This suggests the potential long-term benefits of RPM in managing chronic conditions such as HTN. Moreover, it was observed that the improvements in BP control increased with a greater patient transmission index, indicating the positive impact of regular data transmission on blood pressure management. Future research will include control groups and the effects on patient-reported outcomes and cost reductions.

Given the remote aspect of RPM and its inclusion in a majority of medical coverage plans, this mode of healthcare delivery offers a promising solution to tackle the widespread issue of healthcare accessibility and equity. As the landscape of digital and remote healthcare continues to evolve, it becomes increasingly important to conduct further research into these strategies and optimize them to maximize the influence on population health. Ultimately, by leveraging technology and fostering patient engagement and education, we have the potential to revolutionize the management of chronic conditions, diminish healthcare disparities, and enhance patient outcomes.

## Figures and Tables

**Figure 1 healthcare-12-01583-f001:**
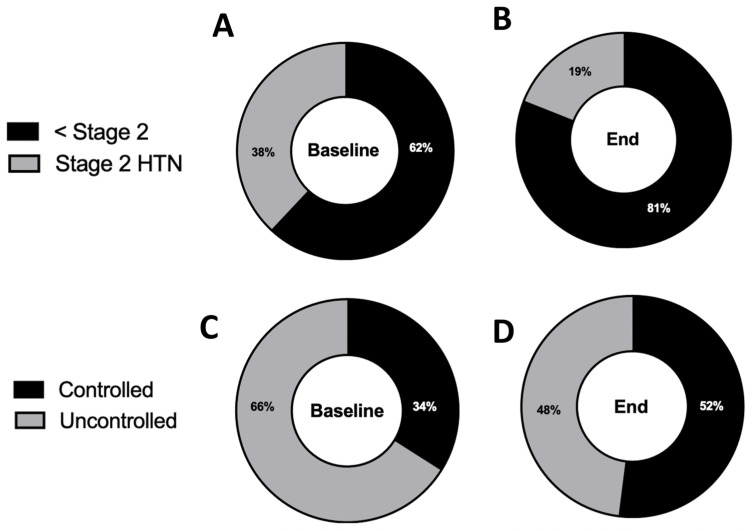
RPM is associated with a reduction in patients with uncontrolled hypertension and stage 2 hypertension. (**A**) Percentage of patients with uncontrolled hypertension at baseline (SBP > 130 mmHg or DBP > 80 mmHg). (**B**) Percentage of patients with uncontrolled hypertension after enrollment in the RPM program. (*n* = 6595, *p* < 0.01, two-tailed McNemar test; average time on RPM = 289 days) (**C**) Percentage of patients with stage 2 hypertension (SBP > 140 mmHg or DBP > 90 mmHg) at baseline. (**D**) Percentage of patients with stage 2 hypertension after enrollment in the RPM program. (*n* = 6595, *p* < 0.01, two-tailed McNemar Test; average time on RPM = 289 days).

**Figure 2 healthcare-12-01583-f002:**
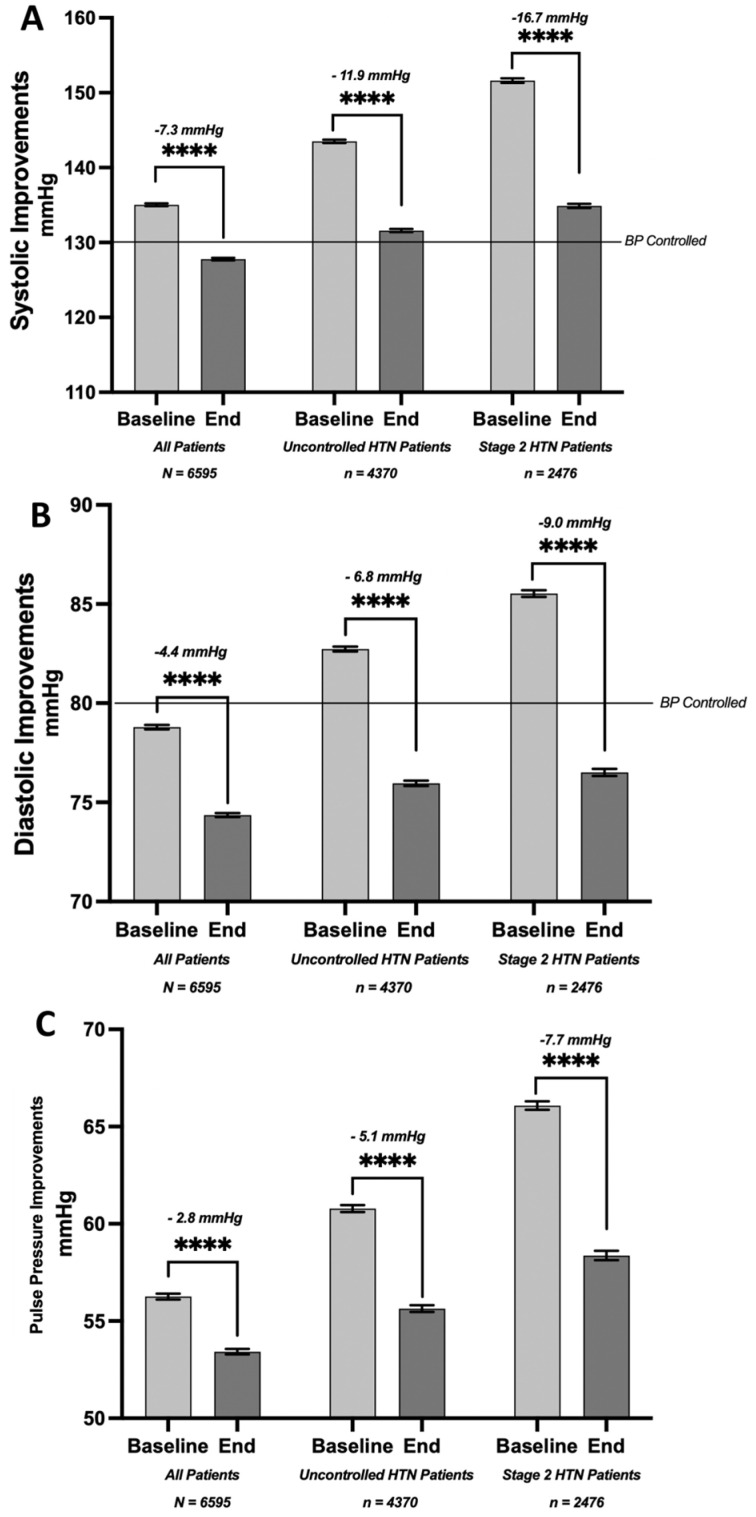
Improvements in systolic and diastolic blood pressure and pulse pressure associated with rpm. (**A**) Change in average systolic blood pressure (in mmHg) from baseline to end of analysis period (mean of 289 days) among all patients (−7.3 mmHg), uncontrolled hypertensives (−11.9 mmHg), and stage 2 hypertensives (−16.7 mmHg). (**B**) Change in average diastolic blood pressure (in mmHg) from baseline to end of analysis period (mean of 289 days) among all patients (−4.4 mmHg), uncontrolled hypertensives (−6.8 mmHg), and stage 2 hypertensives (−9.0 mmHg). (**C**) Change in average pulse pressure (in mmHg) from baseline to end of analysis period (mean of 289 days) among all patients (−4.4 mmHg), uncontrolled hypertensives (−6.8 mmHg), and stage 2 hypertensives (−9.0 mmHg) (paired *t*-test, **** *p* < 0.0001).

**Figure 3 healthcare-12-01583-f003:**
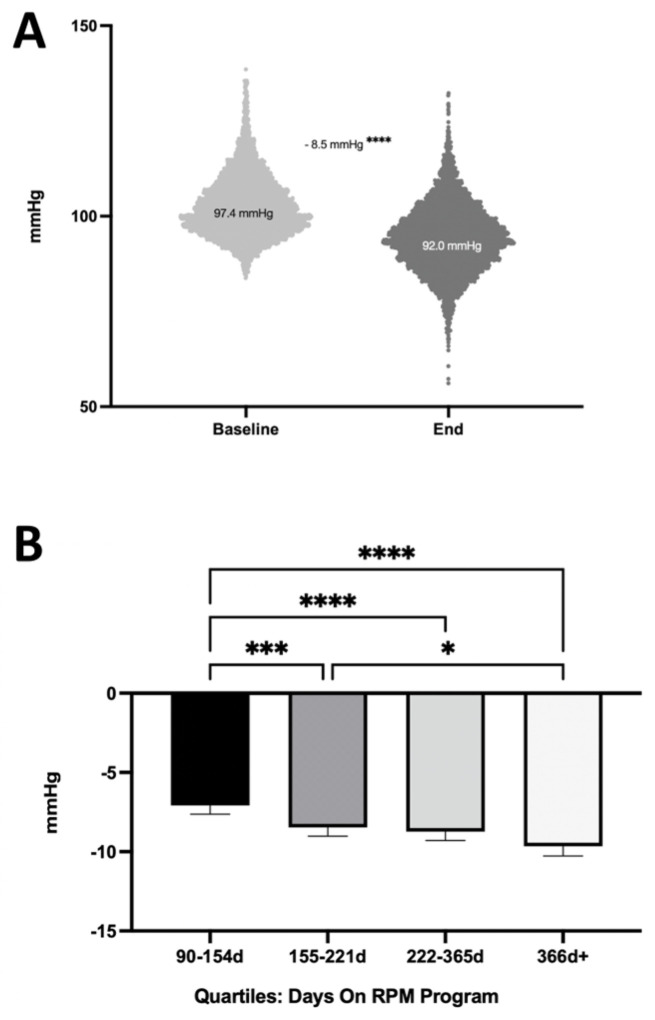
Improvements in mean arterial pressure (MAP). (**A**) Change in average MAP (mmHg) from baseline to end of analysis period among all patients with uncontrolled hypertension (paired *t*-test). (**B**) Improvements in MAP in uncontrolled hypertension over time quartiles of time on the RPM program. More time is associated with a greater improvement (Kruskal–Wallis Test; * *p* < 0.05, *** *p* < 0.001, **** *p* < 0.0001).

## Data Availability

The data presented in this study are available on request from the corresponding author.

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
