# Peer review of "Remote Patient Monitoring Is Associated with Improved Outcomes in Hypertension: A Large, Retrospective, Cohort Analysis"

_healthcare, 2024, doi:10.3390/healthcare12161583_

Round 1

Reviewer 1 Report

Comments and Suggestions for Authors

A nice tidy paper, well done. 

Minor comment(s):

Abstract

Line 14: RPM is not previously define

Line 14: sent blood pressure monitors (not cuffs)

Discussion

Line 233-234: "Research suggests that RPM FOR htn is more cost-effective" needs a reference or two to support this statement.

Main comment:

Based on my understanding of the paper and what was mentioned in the discussion (lines 226-230), there needs to be more clarification in the methods section regarding the patient group. If the patients were already being treated by conventional BP treatment approaches and their prescribed medications did not change then yes it is is fair to say that RPM did improve BP. However, if these are newly diagnosed patients, or have received new medications, etc, it becomes difficult to assess whether the RPM was the underlying component that reduced the individuals BP, which would mean that this study does lose some weight as it becomes more of a RPM feasability study. I say this because in the instance that the individuals did follow a convetional treatment plan would we have seen the same reductions in BP regardless? It would have been nice to have a group that recieved conventional BP treatment so that there was a reference comparator. Perhaps authors could include some references in the dicsussion that show minimal reductions in BP when following a conventional treatment plan compared to when using other RPMs. 

Author Response

Comment 1: Line 14: RPM is not previously define

Response 1: We have written it our here

Comment 2: Line 14: sent blood pressure monitors (not cuffs)

Response 2: We changed it to say "monitor"

Comment 3: Line 233-234: "Research suggests that RPM FOR htn is more cost-effective" needs a reference or two to support this statement.

Response 3: We have added references "De Guzman, K.R., Snoswell, C.L., Taylor, M.L., Gray, L.C. and Caffery, L.J., 2022. Economic evaluations of remote patient monitoring for chronic disease: a systematic review. Value in Health25(6), pp.897-913."

Comment 4: Based on my understanding of the paper and what was mentioned in the discussion (lines 226-230), there needs to be more clarification in the methods section regarding the patient group. If the patients were already being treated by conventional BP treatment approaches and their prescribed medications did not change then yes it is is fair to say that RPM did improve BP. However, if these are newly diagnosed patients, or have received new medications, etc, it becomes difficult to assess whether the RPM was the underlying component that reduced the individuals BP, which would mean that this study does lose some weight as it becomes more of a RPM feasability study. I say this because in the instance that the individuals did follow a convetional treatment plan would we have seen the same reductions in BP regardless? It would have been nice to have a group that recieved conventional BP treatment so that there was a reference comparator. Perhaps authors could include some references in the dicsussion that show minimal reductions in BP when following a conventional treatment plan compared to when using other RPMs. 

Response 4: Thank you for this point. Participants in this study had previously been on conventional HTN treatment for at least 1 year prior to using RPM. We have added this to the methods and discussion sections.

Reviewer 2 Report

Comments and Suggestions for Authors I reviewed the article titled "Remote Patient Monitoring is Associated with Improved Outcomes in Hypertension: a Large, Retrospective, Cohort Analysis” The article title and subtitle are well-chosen and concise. Moreover, the overall organization and presentation very good; however, some further polishing is still necessary to make the article as engaging as possible and of broad immediate and future value.

1. The abstract does not correlate with the content of the article. Please rewrite the abstract and try to make it more informative.

2. The authors are considering improvements to their methodology for remote patient monitoring in hypertension, However, it is not clear how the specific method used in the proposed scheme improves the performance. Authors need to identify additional controls to ensure a more accurate and reliable outcome.

3. There are many typos and grammatical mistakes in the writing. Also,  the article must be checked for linguistic and grammatical errors. Please clear all the issues before resubmitting.

4. The graphs and tables need to be clearly defined and stated. The authors should elaborate on the proof so that it is easily understandable for readers. All the figures and tables must be properly cited in the paper.

5. The results presented from the methodological approach are uninterpreted, and there is no explicit discussion of how they were derived or their significance, making them appear vague. 

6. The problem definition is explained clearly, but a still more relevant explanation is needed that the author could refer to and cite recent papers to enhance your background, i.e., a. Acharya, M., Ali, M. M., Bogulski, C. A., Pandit, A. A., Mahashabde, R. V., Eswaran, H., & Hayes, C. J. (2023). Association of Remote Patient Monitoring with Mortality and Healthcare Utilization in Hypertensive Patients: a Medicare Claims–Based Study. Journal of General Internal Medicine, 1-12. b. Mehmood, G., Khan, M. Z., Bashir, A. K., Al-Otaibi, Y. D., & Khan, S. (2023). An efficient qos-based multi-path routing scheme for smart healthcare monitoring in wireless body area networks. Computers and Electrical Engineering, 109, 108517.

7.  The conclusion section should be enriched with a  brief sentence describing future research prospects.

8. All the references must be uniform and according to the journal format.

Comments on the Quality of English Language

Comments are given

Author Response

Comment 1: The abstract does not correlate with the content of the article. Please rewrite the abstract and try to make it more informative.

Response 1: The abstract has been updated. 

Comment 2: The authors are considering improvements to their methodology for remote patient monitoring in hypertension, However, it is not clear how the specific method used in the proposed scheme improves the performance. Authors need to identify additional controls to ensure a more accurate and reliable outcome.

Response 2: participants in this study had been on conventional HTN treatment for at least a year prior to enrollment in RPM and the baseline time point (this has been added to the methods and discussion sections). While this is still a retrospective cohort analysis, the fact that they had been on conventional therapy already controls for some of the variables mentioned in the limitations section. We are are currently executing a study that includes a a control group of individuals who were prescribed RPM but did not use it (see discussion). 

Comment 3: There are many typos and grammatical mistakes in the writing. Also,  the article must be checked for linguistic and grammatical errors. Please clear all the issues before resubmitting.

Response 3: Thank you for the comment. We have reviewed the manuscript with an outside grammatical editor and made improvements. If there are particular instances that we have missed, please let us know the line numbers.

Comment 4: The graphs and tables need to be clearly defined and stated. The authors should elaborate on the proof so that it is easily understandable for readers. All the figures and tables must be properly cited in the paper.

Response 4: The figure legends have been update to be more descriptive. 

Comment 5: The results presented from the methodological approach are uninterpreted, and there is no explicit discussion of how they were derived or their significance, making them appear vague. 

Response 5: All the results are discussed fully in the discussion section. We report only the facts of the results in the Results section. The subheading of each section in the results indicates the value of the data collected. 

Comment 6: The problem definition is explained clearly, but a still more relevant explanation is needed that the author could refer to and cite recent papers to enhance your background, i.e., a. Acharya, M., Ali, M. M., Bogulski, C. A., Pandit, A. A., Mahashabde, R. V., Eswaran, H., & Hayes, C. J. (2023). Association of Remote Patient Monitoring with Mortality and Healthcare Utilization in Hypertensive Patients: a Medicare Claims–Based Study. Journal of General Internal Medicine, 1-12. b. Mehmood, G., Khan, M. Z., Bashir, A. K., Al-Otaibi, Y. D., & Khan, S. (2023). An efficient qos-based multi-path routing scheme for smart healthcare monitoring in wireless body area networks. Computers and Electrical Engineering, 109, 108517.

Response 6: These citations have been added in line 233

Comment 7: The conclusion section should be enriched with a  brief sentence describing future research prospects.

Response 7: This has been added "Future research will include control groups and the effects on patient reported outcomes and cost reductions."

Comment 8: All the references must be uniform and according to the journal format.

Response 8: Citations have been updated.

Reviewer 3 Report

Comments and Suggestions for Authors

Dear Authors,

The manuscript "Remote Patient Monitoring is Associated with Improved Outcomes in Hypertension: a Large, Retrospective, Cohort Analysis" is very interesting. Analysis is robust with the big cohort size. However, I have 2 comments.

1. Were there any other exclusion criteria for participation ?

2. Wouldn't it be interesting to know if certain age, gender,or race benefit the most from RPM?  Could you provide demographic summary of the participants ?

Best.

Author Response

Comment 1: Were there any other exclusion criteria for participation ?

Response 1: All inclusion and exclusion criteria are listed in the methods section (especially lines 90-97 for exclusion criteria). All participants were treated with conventional HTN management for at least a year prior to RPM enrollment (this has been added in line 70).

Comment 2: Wouldn't it be interesting to know if certain age, gender,or race benefit the most from RPM?  Could you provide demographic summary of the participants ?

Response 2: We agree that demographics would be interesting for the analysis. Unfortunately, we do not have access to demographics in this data set due to data privacy and data use agreements at the time of this study. We only have access to deidentified and pooled data. Several ongoing studies have modified these data use agreements and IRB protocols to allow for specific demographics, this is an open area of ongoing work. 

Reviewer 4 Report

Comments and Suggestions for Authors

Thank you for providing me the opportunity to review the paper entitled " Remote Patient Monitoring is Associated with Improved Outcomes in Hypertension: A Large, Retrospective, Cohort Analysis ".

General comments:

This study aimed to evaluate the association between RPM and changes in blood pressure in the HTN population. The authors concluded that there was improvement in MAP in all patients with Uncontrolled HTN by 8.5 mmHg (p<0.0001). RPM is associated with improved HTN control and provides further evidence supporting telehealth programs which can aid in chronic disease management.  It is a well-written manuscript, and the findings are well presented. I have some comments.

Minor

1. There’s no baseline characteristics in both groups. Please add on those. Because, I just wonder if there would be similar baseline characteristics in both groups

2. also please add those medications in both groups.

3. please clarify when the patients did check blood pressure

Author Response

comment 1: There’s no baseline characteristics in both groups. Please add on those. Because, I just wonder if there would be similar baseline characteristics in both groups

Response 1: We agree that demographics would be interesting for the analysis. Unfortunately, we do not have access to demographics in this data set due to data privacy and data use agreements at the time of this study. We only have access to deidentified and pooled data. Several ongoing studies have modified these data use agreements and IRB protocols to allow for demographics, this is an open area of ongoing work. 

Comment 2: also please add those medications in both groups.

Response 2: As stated in Response 1, we do not have access to medication information in this data set. We are actively collecting it for a follow up study we are working on in a different population and will correlate the results when we have them.

Comment 3: please clarify when the patients did check blood pressure. 

Response 3: as stated in the methods, "Upon enrolling in the RPM program, prescribed patients received a cellular-enabled, FDA-cleared, digital, automatic blood pressure monitor from HealthSnap, Inc. This device automatically transmitted data to a HIPAA-compliant remote patient portal. A team of HealthSnap licensed care navigators (registered nurses) monitored this portal, guiding patients on how to correctly measure their blood pressure according to the American Heart Association's recommended protocol [13]. Briefly, patients were advised to follow certain pre-measurement steps including resting quietly, avoiding caffeine, exercise, or smoking, emptying their bladder, and wearing comfortable clothing. They were also educated on the correct sitting position and cuff placement for accurate readings. To account for variations and confirm reading accuracy, two consecutive measurements were taken for each participant, with a 1–2-minute interval between each measurement."

Participants could take their blood pressure readings at any time of day and multiple times per day. to account for time and frequency variability, the entire week's readings were averaged together to obtain the weekly value that we used for our analysis. this has been added to the methods section. thank you for the comment.

Round 2

Reviewer 1 Report

Comments and Suggestions for Authors

No further comments. Nice work with the manuscript. 

Reviewer 2 Report

Comments and Suggestions for Authors

The author addressed my concern, so i accept the paper.

Comments on the Quality of English Language

Nill

Reviewer 3 Report

Comments and Suggestions for Authors

Dear Authors,

I have no further comments.

Best.

Comments on the Quality of English Language

Minor editing of English language required